# Inhibition of Heme Oxygenase-1 Activity Enhances Wilms Tumor-1-Specific T-Cell Responses in Cancer Immunotherapy

**DOI:** 10.3390/ijms20030482

**Published:** 2019-01-23

**Authors:** Damaris A. Schillingmann, Sebastian B. Riese, Vijith Vijayan, Sabine Tischer-Zimmermann, Helga Schmetzer, Britta Maecker-Kolhoff, Rainer Blasczyk, Stephan Immenschuh, Britta Eiz-Vesper

**Affiliations:** 1Institute for Transfusion Medicine, Hannover Medical School, 30625 Hannover, Germany; damaris.schillingmann@googlemail.com (D.A.S.); riese.sebastian@mh-hannover.de (S.B.R.); vijayan.vijith@mh-hannover.de (V.V.); tischer-zimmermann.sabine@mh-hannover.de (S.T.-Z.); blasczyk.rainer@mh-hannover.de (R.B.); immenschuh.stephan@mh-hannover.de (S.I.); 2Integrated Research and Treatment Center (IFB-Tx), Hannover Medical School, 30625 Hannover, Germany; maecker-kolhoff.britta@mh-hannover.de; 3Department for Hematopoetic Transplantations, Med. III, University Hospital Munich-Grosshadern, 81377 Munich, Germany; helga.schmetzer@med.uni-muenchen.de; 4Department of Pediatric Hematology and Oncology, Hannover Medical School, 30625 Hannover, Germany

**Keywords:** Antigen-specific T cells, heme oxygenase 1 (HO-1) inhibition, immunotherapy, Wilms tumor protein-1 (WT1), acute myeloid leukemia (AML)

## Abstract

Wilms tumor protein-1 (WT1) is an attractive target for adoptive T-cell therapy due to its expression in solid tumors and hematologic malignancies. However, T cells recognizing WT1 occur in low frequencies in the peripheral blood of healthy donors, limiting potential therapeutic possibilities. Tin mesoporphyrin (SnMP) is known to inhibit heme oxygenase-1 (HO-1), which has been shown to boost the activation and proliferation of human virus-specific T cells. We analyzed the influence of this effect on the generation of WT1-specific T cells and developed strategies for generating quantities of these cells from healthy donors, sufficient for adoptive T-cell therapies. HO-1 inhibition with SnMP increased WT1-specific T-cell frequencies in 13 (26%) of 50 healthy donors. To assess clinical applicability, we measured the enrichment efficiency of SnMP-treated WT1-specific T cells in response to a WT1-specific peptide pool and a HLA-A*02:01-restricted WT1 peptide by cytokine secretion assay. SnMP treatment resulted in a 28-fold higher enrichment efficacy with equal functionality. In conclusion, pharmacological inhibition of HO-1 activity with SnMP results in more efficient generation of functionally active WT1-specific T cells. This study demonstrates the therapeutic potentials of inhibiting HO-1 with SnMP to enhance antigen-specific T-cell responses in the treatment of cancer patients with WT1-positive disease.

## 1. Introduction

Decreased relapse in leukemia patients after hematopoietic stem cell transplantation (HSCT) with non-T-cell-depleted allografts versus depleted allografts [1] may be attributed to the graft-versus-leukemia (GvL) potential of T lymphocytes. Tumor-associated antigens (TAAs) such as the Wilms tumor gene WT1 have the potential to generate tumor antigen-specific T cells which promote a GvL effect and display selective cytotoxicity to leukemic cells [2,3,4] while avoiding side effects of allogeneic HSCT like graft versus host disease (GvHD). WT1 is a zinc-finger transcription factor expressed during normal ontogenesis. WT1 expression is absent or very low in healthy adult tissue, but abundant in a wide range of malignancies and solid tumors, including lung, breast, and colon cancer and soft tissue sarcomas [5,6,7,8]. Likewise, WT1 is overexpressed in 60 to 100% of patients with acute lymphoblastic leukemia (ALL) or acute myeloid leukemia (AML) [9].

Evidence shows that a decrease in circulating malignant cells is strongly associated with a decrease in WT1-encoding mRNA levels as well as a reduced risk of relapse and prolonged disease-free survival [10,11]. This makes WT1 an attractive target for adoptive immunotherapy. CD8^+^ cytotoxic T-lymphocytes (CTLs) specific for WT1 occur in cancer patients as well as in low frequencies in healthy donors [10,12].

Weber and colleagues generated autologous T-cell lines from ALL patients by targeting the tumor-associated antigens WT1, MAGE-A3, Survivin, and PRAME [13], which were able to specifically kill autologous leukemia blasts in vitro. They detected responses for WT1-specific CD4^+^ T helper cells as well as CD8^+^ CTLs, indicating that the combination of CD8^+^- and CD4^+^-restricted epitopes might improve T-cell activation and survival, and may facilitate a better anti-leukemic effect in vivo [13,14,15]. The detection and generation of tumor-specific T cells from peripheral blood of healthy donors is feasible [16,17], but the ability to expand these cells in vitro is limited due to the low frequency of memory precursor WT1-specific T cells present in healthy donors. Potent mediators of T-cell proliferation are needed to generate WT1-specific T cells from healthy donors in numbers sufficient for adoptive immunotherapy. Heme oxygenase-1 (HO-1) is a promising therapeutic target for the modulation of T-cell responses. HO-1, the inducible isoform of the enzyme heme oxygenase, also known as heat shock protein 32 (HSP32), catalyzes the first and rate-limiting step from heme to biliverdin, yielding carbon monoxide (CO) and free iron. HO-1 and its byproducts have been found to mediate immunomodulatory effects by antioxidant, anti-proliferative, anti-inflammatory, anti-apoptotic, and immunosuppressive mechanisms in various cell types [18,19,20,21]. Pae et al. found that T-cell activation increases HO-1 activity and CO production from heme, which exerts an antiproliferative effect on T cells by suppressing IL-2 secretion [22]. HO-1 inhibition with the metalloporphyrin SnMP, on the other hand, induces the antigen-independent activation, proliferation and maturation of naïve CD4^+^ and CD8^+^ T cells in vitro [18]. Our recent study showed that SnMP-mediated inhibition of HO-1 activity significantly increases the number of cytomegalovirus (CMV)-specific T cells and promotes a more active phenotype suggesting that HO-1 is a promising therapeutic target for the modulation of memory T-cell responses and the clinical application of T cells [23].

The two main aims of the present study were (1) to determine if the number of functionally active WT1-specific T cells obtained from healthy donors can be increased in vitro by inhibiting the enzymatic activity of HO-1 and (2) to examine whether HO-1 inhibition makes it possible to generate sufficient cell numbers for adoptive T-cell transfer to effectively augment T-cell immunity in leukemia patients. The results of this study should help extend the applicability of T-cell therapy to the majority of patients. To this end, we evaluated the frequencies and phenotypes of specific T cells before and after HO-1 inhibition with SnMP and looked for potential effects of HO-1 inhibition on signaling by flow cytometry. Furthermore, we examined the possibility of enriching WT1-specific T cells in a near-clinical setting and analyzed the effects of HO-1 on cells activated antigen-dependently and -independently by flow cytometry, ELISpot and ELISA. Our findings highlight the promising therapeutic potential of strategies devised to up- or downregulate HO-1 activity.

## 2. Results

### 2.1. HO-1 Inhibition Had No Significant Effect on T-Cell Activation and Subsets in an Antigen-Independent Setting

To assess the effect of inhibiting the enzymatic activity of HO-1 via SnMP, CD3^+^ T cells (*n* = 7) were stimulated in an antigen-independent manner for 6 days with CD3/CD28 beads. Phenotype analysis of the CD3^+^, CD4^+^, and CD8^+^ T cells revealed time-dependent changes. T_N_ and T_EMRA_ cell counts increased on the first day, but decreased dramatically after six days. In contrast, the numbers of T_CM_ and T_EM_ were higher on day 6 than on day 0, but stimulation with SnMP did not lead to significant alteration of the T-cell phenotype in the CD3^+^, CD8^+^, and CD4^+^ T-cell populations (Figure 1A). 

SnMP had no significant effect on the expression of programmed cell death receptor-1 (PD-1) in CD3^+^, CD8^+^ and CD4^+^ T-cell populations. The highest PD-1 expression levels were found on day 3: 39.4% in CD4^+^, 27.1% in CD3^+^, and 24.7% in CD8^+^ SnMP-untreated T cells. PD-1 expression in SnMP-treated cells was 3% to 6% lower than in SnMP-untreated cells (Figure 1B). As expected, analysis of IFN-γ on transcriptional level showed the highest amount of IFN-γ mRNA on day 1 in cells treated with and without SnMP. The highest amounts of miRNA-155 were observed on day 2 in SnMP-treated cells and on day 3 in SnMP-untreated cells. Nevertheless, the differences between cells treated with and without SnMP were not significant at either the miRNA-155 level or the IFN-γ mRNA level (Figure 1C).

As determined by ELISA, the highest concentrations of granzyme B (+ SnMP: 135.99 ng/mL, − SnMP: 135.87 ng/mL), and IFN-γ (+ SnMP: 59.63 ng/mL, − SnMP: 75.96 ng/mL), respectively, were detected on days 0, 2, 3, and 6 (data shown only for days 0 and 6). HO-1 inhibition with SnMP did not significantly alter the secretion level of the effector molecules (Figure 1D).

### 2.2. SnMP Resulted in Higher T-Cell Response to WT1 in Healthy Donors

To demonstrate the antigen-dependent effects of HO-1 inhibition, peripheral blood mononuclear cells (PBMCs) from healthy donors were treated with or without SnMP, stimulated with an overlapping pool of peptides derived from WT1 (ppWT1), and analyzed by IFN-γ ELISpot. HO-1 inhibition with SnMP led to a significant (30.1-fold) increase in the number of IFN-γ-specific spots (21.1 spots per 2.5 × 10^5^ cells) compared to cells stimulated without SnMP (0.7 spots per 2.5 × 10^5^ cells) (Figure 2A and Appendix A). Analysis of DMSO-treated (solvent control) and untreated cells showed no significant differences (data not shown) compared to non-stimulated cells.

### 2.3. Combined Stimulation with ppWT1 + SnMP Enhanced the Enrichment of IFN-γ-Secreting T Cells in a Preclinical Setting

To address the question of whether it might be feasible to use SnMP to enrich tumor-specific T cells in a clinical environment, we used the IFN-γ cytokine secretion assay (CSA) comparable to the cytokine capture system (CCS) used for the enrichment of clinical grade T cells for adoptive T-cell transfer. Prior to enrichment, the CD3^+^/IFN-γ^+^ T-cell response to ppWT1 was 1.2-fold higher in cells treated with ppWT1+SnMP (0.16%) than in those stimulated with ppWT1 alone (0.14%). After enrichment, the amount of CD3^+^/IFN-γ^+^ T cells obtained was significantly higher (2.7-fold) in the experimental set-up with ppWT1 + SnMP (52.0%) compared to ppWT1 alone (19.0%) (Figure 2B, left).

For direct comparison of antiviral and WT1-specific T-cell response, PBMCs from four healthy donors were stimulated with ppEBV Consensus in the presence and absence of ppWT1. Before enrichment, the CD3^+^/IFN-γ^+^ T-cell response to ppEBV Consensus was slightly higher (1.2 fold) in in the presence of SnMP (0.5%) than in its absence (0.43%). After enrichment, the CD3^+^/IFN-γ^+^ T-cell response was 1.2-fold higher in cells treated with ppEBV+SnMP (44.2%) than in those treated with ppEBV alone (36.1%) (Figure 2B). After enrichment of IFN-γ-positive T cells by MACS, the highest purity was achieved in the experimental set-up with SnMP + ppWT1 (85.3% CD3^+^/IFN-γ^+^ T cells).

### 2.4. SnMP Increases Phosphorylated ERK Expression

To demonstrate the effect of SnMP on the T-cell receptor (TCR) signaling pathway, PBMCs from six healthy donors were stimulated using ppWT1 with or without SnMP, and the percentage of expressed pERK1/2 in CD3^+^, CD4^+^, and CD8^+^ T-cell populations was assessed by flow cytometry. All cell populations expressed higher amounts of pERK1/2 after treatment with ppWT1+SnMP compared to ppWT1 alone (Figure 3A and Appendix A). The amount of pERK increased significantly in response to HO-1 inhibition: CD3^+^ T cells showed a 2-fold increase from 0.8% (ppWT1) to 1.6% (ppWT1 + SnMP), CD4^+^ T cells a 1.6-fold increase from 0.9% (ppWT1) to 1.4% (ppWT1 + SnMP), and CD8^+^ T cells a significant 2.4-fold increase, from 0.7% to 1.7%. These findings indicate that HO-1 inhibitor SnMP was responsible for the increased activation of these signaling pathways important for T-cell activation.

### 2.5. CD137 Expression Increases in Response to SnMP

CD3^+^, CD4^+^, and CD8^+^ T-cell populations showed higher CD137 expression when stimulated with ppWT1+SnMP (CD3^+^: 1.9%, CD4^+^: 1.5%, CD8^+^: 13.1%) than with ppWT1 alone (CD3^+^: 0.4%, CD4^+^: 0.2%, CD8^+^: 2.3%). This means that SnMP increased CD137 expression 4.8-fold in CD3^+^ T cells, 7.5-fold in CD4^+^ T cells, and 5.7-fold in CD8^+^ T cells. In the case of CD4^+^ T cells, the difference was significant (Figure 3B).

### 2.6. Effect of SnMP on WT1 Peptide-Specific T-Cell Expansion and T-cell Functionality

To demonstrate the effect of SnMP on WT1 peptide-specific T-cell expansion, PBMCs were stimulated with pWT1_37–45_ peptides with or without SnMP for 24 h, after which CD137^+^ cells were isolated and cultured with pWT1_37–45_-loaded artificial antigen-presenting cells (aAPCs) for seven days. The numbers of CD137^+^ cells on days 0 and 7 were higher in cells treated with pWT1_37–45_ + SnMP than in those treated with pWT1_37–45_ alone (1.4-fold and 3.1-fold increase, respectively) (Figure 4A). 

The composition of T-cell subsets was assessed and there was no significant difference between the groups treated with or without SnMP in the CD3^+^, CD8^+^ and antigen-specific HLA-A*02:01-restricted pWT1_37–45_ populations (Figure 4B).

Treatment of cells from HLA-A*02:01-positive donors with the HLA-A*02:01 restricted pWT1_37–45_ peptide and SnMP led to an 5.3-fold increase in CD137 expression in CD3^+^ T cells compared to the levels achieved with pWT1_37–45_ alone (0.3% to 1.6%). In CD4^+^ T cells, the expression of CD137 after treatment with pWT1_37–45_ + SnMP (1.3%) was 13-fold higher than with pWT1_37–45_ alone (0.1%). In both T-cell populations, the difference was significant. In CD8^+^ T cells, the increase in CD137 expression was 6.7-fold (pWT1_37–45_ peptide: 1.1%, pWT1_37–45_ + SnMP: 7.4%). The same effect was observed in A*02 WT1_37–45_ positive cells alone, which showed a 3-fold increase after treatment with pWT1_37–45_+SnMP (21.1%) compared to pWT1_37–45_ alone (7.1%) (Figure 5A and Appendix A).

In order to test the specificity and functionality of expanded cells, we performed IFN-γ and granzyme B ELISpots and ELISAs on the seventh day. HLA-A*02-expressing K562 cells loaded with pWT1_37−45_ served as the target cells, which were incubated with the expanded cells. Supernatants were collected and their IFN-γ and granzyme B concentrations were determined by ELISA. The results showed a comparable concentration of IFN-γ in the SnMP-treated and -untreated groups. Nevertheless, the SnMP-treated cells had higher concentrations of granzyme B in the supernatant than in the SnMP-untreated cells, but the differences were not significant (Figure 5B).

When considering specifically the antigen-specific pWT1_37–45_ T cells a higher number of spots per 1000 cells in both granzyme B (Figure 5C) and IFN-γ ELISpot assay (Figure 5D) compared to analysis of the whole CD3^+^ T-cell population were determined. For both cytokines, the spot counts in the SnMP-treated groups were higher than those in the SnMP-untreated groups: the corresponding increases were 2.4-fold (ratio of 0.1:1), 1.5-fold (ratio of 0.05:1) and 2.1-fold (ratio of 0.01:1) for granzyme B, and 1.2-fold (ratios of 0.1:1 and 0.05:1) and 1.6-fold (ratio of 0.01:1) for IFN-γ.

## 3. Discussion

Previously, we observed that the pharmacologic agent SnMP inhibits the enzymatic activity of HO-1, resulting in the increased activation and proliferation of human virus-specific T cells [23]. The present study investigated the impact of SnMP-based HO-1 inhibition on the generation of T cells specific for the tumor antigen WT1. Wilms tumor protein-1 has been described as a promising target for immunotherapeutic approaches for various reasons [24,25], including its role in the survival of cancer cells [26,27]. The WT1 gene acts as an oncogenic factor rather than as a tumor suppressor [28], and is involved in blocking the further differentiation of hematopoietic progenitor cells.

Our results indicate that HO-1 inhibition results in a significant increase in WT1-specific T cells. The findings of this study will contribute to the development of strategies for generating sufficient quantities of WT1 leukemia-specific T cells from healthy donors to augment effective T-cell immunity in leukemia patients and to broaden the applicability of T-cell-based therapies for leukemia.

### 3.1. HO-1 Inhibition Improves Clinical Enrichment Strategies for WT1-Specific T Cells

In order to mimic a clinical setting, we used the CSA because it is comparable to the CCS, a technology used for the enrichment of antigen-specific T cells for adoptive T-cell transfer in clinical applications [29,30,31]. After integrating SnMP into the enrichment process, we showed that the use of this mesoporphyrin makes the generation of tumor-specific T cells feasible on a larger scale. 

Our results confirm the beneficial effect of HO-1 inhibition on the enrichment procedure itself. Thus, the use of HO-1 inhibitors might make it possible to effectively enrich WT1-specific T cells despite their low frequencies in the peripheral blood of healthy donors. 

The determined ELISpot assay data concurred with those of other studies [32] on the occurrence of WT1-specific T cells in healthy donors. In our study, roughly one-quarter of the 50 healthy donors examined showed a positive response to ppWT1. 

It remains partly unclear why WT1-specific T cells are not only present in cancer patients, but also in healthy individuals. According to Schmied et al., WT1-specific T cells are not negatively selected in the thymus [17]. Pinto et al. demonstrated that, in the case of melanoma antigen recognized by T cells 1 (MART-1), truncated mRNA isoforms of the MART-1 epitope are expressed in human medullary thymic epithelial cells, leading to partial evasion of self-tolerance mechanisms towards this epitope [33]. In addition, Steger et al. could correlate increased proportions of leukemia-specific cells with prolonged relapse-free survival after stem cell transplantation [34]. Rezvani et al., found that leukemia-specific T cells of healthy donors showed a diversity concerning their functional avidity towards the antigen [32,35]. Notably, in leukemia patients a significantly higher number of antigen-specific T cells with low-avidity was found compared to healthy donors. 

The CSA and T-cell expansion protocols focus on the detection of IFN-γ or expression markers like CD137. In the present study, their expression on CD3^+^, CD4^+^ and CD8^+^ T cells increased after stimulation with ppWT1 plus SnMP compared to stimulation with ppWT1 alone.

Yet another important aspect is the effect of WT1 antigen stimulation plus HO-1 inhibition on an essential intracellular pathway. Activation of the ERK pathway occurs upon T-cell activation and is required for reactions such as T-cell proliferation and the stimulation of IL-2 gene transcription in T cells [36]. Furthermore, the CO produced during the reaction catalyzed by HO-1 has been shown to inhibit this pathway and thereby influence T-cell proliferation [22]. 

Therefore, the flow cytometric assessment of pERK expression addressed the question of whether HO-1 inhibition affects the activation of this cascade. The significant increase in pERK expression in cells stimulated with ppWT1 plus SnMP demonstrates the positive effect of HO-1 inhibition on subcellular pathways important for the proliferation of potential WT1-specific T cells. 

### 3.2. HO-1 Inhibition in the Absence of Antigenic Stimulation Did Not Alter T-Cell-Functionality

Treatment of PBMCs with or without HO-1 inhibitor SnMP in the absence of antigenic stimulation resulted in no significant difference in PD-1 expression. PD-1 is a receptor expressed by activated T cells and B cells [37,38]. Binding to its ligands PD-L1 and PD-L2 leads to the inhibition of T-cell activation [39] and thus plays an important role in the regulation of T-cell responses [40,41]. Hence, researchers have also attempted to use PD-1 as a target for adoptive immunotherapy, for example, in the treatment of melanoma [42]. Our findings suggest that, although a slightly higher percentage of PD-1 seemed to be expressed in the absence of SnMP, HO-1 inhibition did not have a significant effect on the activation of T cells in the absence of antigenic stimulation. This is consistent with the lack of differences in cytokine concentrations (IFN-γ and granzyme B) in the respective supernatants. The same applied to the amount the miRNA-155 and IFN-γ and to the phenotypes of CD3^+^, CD4^+^ and CD8^+^ T cells, which also were not influenced by SnMP. 

Apparently, while SnMP had no effect on the activation of T cells in an antigen-independent manner, it is likely, that there is a low risk of side effects when using SnMP in these indications.

### 3.3. HO-1 Inhibition Increases pWT1_37–45_ Peptide-Specific T-Cell Expansion and Functionality Without Altering the Composition of T-Cell Subsets

Based on the preliminary results, we postulated that using ppWT1 in combination with SnMP would have a positive effect on the enrichment on WT1-specific T cells. In order to test this hypothesis, it was necessary to use HLA-A2-restricted peptides. We chose HLA-A*02:01/VLDFAPPGA (pWT1_37–45_) and HLA-A*02:01/RMFPNAPYL (pWT1_126–134_, data not shown) peptides, as they are from two of the HLA-A*02:01-restricted WT1 epitopes implicated in leukemia. pWT1_126–134_ vaccines for myeloid leukemia have been investigated in several studies [43,44], and pWT1_37–45_-specific CD8^+^ T cells of high avidity have been detected in patients with myeloid leukemia [32]. Furthermore, Schmied et al. achieved promising results in the enrichment of pWT1_37–45_-specific CD8^+^ T cells [17].

After stimulation with the respective peptides in the presence and absence of HO-1 inhibitor SnMP, cells pulsed with both pWT1_37–45_ and pWT_126–134_ peptides (data not shown) showed increased numbers of CD137^+^ cells in response to HO-1 inhibition. We therefore concluded that a higher proliferation rate can be achieved by this method. Although the use of HLA*02:01-restricted peptides can lead to significant T-cell responses, we believe that the use of peptide pools will simplify the production of customized T cells because it offers the possibility for HLA-independent application.

In order to use SnMP in a clinical setting, its activating properties must increase the number of tumor-specific T cells whilst leaving the composition of the T-cell subsets entirely intact. In our experiments, there was no significant difference between the SnMP-treated and -untreated groups in the CD3^+^, CD8^+^ and Multimer^+^ T-cell populations. This is comparable to our earlier findings on HO-1 inhibition with SnMP in anti-viral T cells [23], and to our results achieved using SnMP in an antigen-independent setting.

Our IFN-γ and granzyme B ELISA analysis showed comparable IFN-γ concentrations and higher granzyme B concentrations in the supernatants of SnMP-treated cells compared to the concentrations in SnMP-untreated cells. The IFN-γ and granzyme B ELISpots which we performed to assess the ability of the WT1-peptide pulsed cells to recognize target cells and measure IFN-γ and granzyme B as sign of reaction showed no difference between SnMP-treated and SnMP-untreated CD3^+^ T cells. However, an increase in IFN-γ and granzyme B spots in the SnMP-treated cells of the pWT1_37–45_ dextramer fraction was observed, showing that these cells are indeed more functionally active than those generated without SnMP. 

An important goal in customizing anti-tumor T-cell therapy is to achieve a long lasting GvL effect without increasing the risks of GvHD. In one study, adoptive T-cell transfer with tumor-infiltrating lymphocytes resulted in long-lasting complete responses in metastatic melanoma patients, which were detected even several months after the treatment [45]. 

SnMP has also been used to treat neonatal hyperbilirubinemia patients with a low risk profile. In 2016, Bhutani et al. published a placebo-controlled, multicenter trial on the treatment of neonatal hyperbilirubinemia in which over 200 newborns were included. In this case, no severe short-term effects were described, which is a promising result for the future use of SnMP in a clinical setting [46,47]. Future studies should investigate whether SnMP has a positive effect on the long-term frequency of tumor-specific T cells in leukemia patients. These long-term effects have also been discussed in association with other methods used to induce a GvL effect. The donor lymphocyte infusion (DLI) for example, has shown especially promising results regarding the reduction of relapse rates in CML after HSCT [48]. In patients with AML and ALL, however, response rates remain limited or restricted to a small group of patients [49].

Furthermore, although the risk of GvHD can be influenced significantly by several factors, such as the timing of infusion, the type of disease or donor–recipient HLA-status, the risk of GvHD still remains [50].

Interestingly, DLI in combination with azacitidine, an analog for the nucleoside cytidine, achieved a distinct but not durable WT1-specific response, as described in one case report [51].

As mentioned above, several clinical trials have focused on using vaccines against tumor antigens. In the case of WT1 peptide vaccines, several approaches have confirmed their low risk of side effects and feasibility for the treatment of solid tumors like pancreatic cancer [52] as well as AML and myelodysplastic syndrome (MDS) [43]. Like Di Stasi et al. point out in their review of nine different studies on peptide vaccinations in AML and MDS, the functionality and frequency of WT1-specific T cells improves after the vaccinations [53]. The challenge of achieving a sustainable immune response and expression of WT1 in antigen-presenting cells is being addressed in ongoing research.

Transgenic methods for the very antigen-specific production and modulation of T cells have been established recently. Xue et al. showed that T cells isolated from CML and AML patients and transduced with a WT1-TCR gene are able to eliminate leukemic cells in an HLA-A2-restricted manner in the non-obese diabetic–severe combined immunodeficiency (NOD/SCID) murine model [54]. Another recent clinical trial using retroviral transduction of WT1-specific TCR genes in AML and MDS patients also produced promising results in terms of T-cell survival in those patients in the absence of immediate severe side effects [55]. However, the effects of these therapies on the survival or outcome of these patients have yet to be determined.

Another related focus of recent investigation was the development of so-called chimeric antigen receptor (CAR) T cells. These engineered synthetic receptors consist of single-chain variable fragments, which originate from monoclonal antibodies (mAbs) and are thought to detect a specific antigen, transmembrane region, and intracellular domain involved in intracellular signaling [56,57,58]. These receptors display a higher affinity towards their antigen and recognize it independently from the MHC system, albeit with lower sensitivity than conventional TCRs, which are able to recognize even intracellular antigens via the MHC system [59]. CAR-based therapies have produced promising results in patients with hematologic diseases [60,61,62], but seem to be less successful in solid tumors, where encouraging results are restricted to few tumor types [63].

Whereas certain expected side effects such as insertional oncogenesis did not occur, others like on-target/off-tumor recognition and cytokine release syndrome (CRS) were observed in several cases [64]. In light of this, the use of WT1-specific T cells generated from healthy donors in sufficient numbers to treat solid tumors may well be a convincing and more cost-effective alternative that merits future investigation. 

In summary, we showed that HO-1 inhibition with SnMP is a promising way to increase the activation of WT1-specific T cells to generate more active T cells for adoptive T-cell therapy. We see this approach as a first step towards the large-scale production of WT1-specific T cells. The issue of whether HO-1 inhibition also improves the survival of antigen-specific T cells in vivo and thus allows patients to develop a long-term anti-tumor response remains an open question for future research.

## 4. Material and Methods

### 4.1. Study Population

All experiments were performed with residual blood samples from platelet apheresis disposable kits used for routine platelet collection and from regular anonymous healthy donors. After obtaining written informed consent, peripheral blood was collected from healthy platelet donors at the Hannover Medical School (MHH) Institute for Transfusion Medicine. Healthy donors were typed for HLA class I and class II alleles at the four-digit level by sequence-based subtyping. Peripheral blood mononuclear cells (PBMCs) from healthy thrombocyte donors with no prior history of blood transfusion and no signs of acute infection were isolated using discontinuous gradient centrifugation [65].

### 4.2. Synthetic Peptide Pools and Peptides

Overlapping pools of WT1-derived (ppWT1, Miltenyi Biotec, Bergisch Gladbach, Germany) and EBV-derived peptides (PepTivator EBV Consensus, ppEBV, Miltenyi Biotec) were used to stimulate antigen-specific memory T cells, and HLA-A*02-restricted WT1-derived peptides VLDFAPPGA (pWT1_37–45_, EZBiolab, Carmel, IN, USA) and RMFPNAPYL (pWT1_126–134_, EZBiolab) were used to stimulate WT1-specific memory T cells.

### 4.3. Influence of SnMP in an Antigen-Independent Setting

To determine the effect of SnMP (PeproTech, Hamburg, Germany) treatment regarding functionality and phenotype on T-cells in an antigen-independent setting, PBMCs from healthy donors (*n* = 7) were co-cultured with 10 µM SnMP in the absence of WT1-specific antigens. Isolated CD3^+^ T cells (human Pan T cell Isolation Kit (Miltenyi Biotec) from same donors were co-cultured with anti-CD3/CD28 beads (Dynabeads^®^ Human T-Activator CD3/28, ThermoFisher Scientific, Schwerte, Germany) according to the manufacturer’s instructions in T-Cell Medium 1 (TCM1)-composed of RPMI 1640 medium (Lonza, Verviers, Belgium) with 10% human AB serum (C.C. pro, Oberdorla, Germany) and 50 U/mL human IL-2 (PeproTech, Hamburg, Germany) for six days in the presence and absence of SnMP. Aliquots of cells were collected on days 1, 2, 3, and 6 for T-cell phenotype and PD-1 expression analysis (anti-PD-1-phycoerythrin (PE), BioLegend, London, Great Britain) by multicolor flow cytometry, and for IFN-γ and miRNA-155 gene expression analysis by quantitative real-time PCR.

### 4.4. Evaluation of Changes the T-Cell Frequencies and Phenotypes 

Specific T-cell phenotype and frequencies was assessed after treatment with 10 µM SnMP in an antigen-independent setting as well as before and after expansion of WT1-specific T cells. To distinguish between naïve T cells (T_N_, CD62L^+^ CD45RA^+^), central memory T cells (T_CM_, CD62L^+^ CD45RA^−^), effector memory T cells (T_EM_, CD62L^−^ CD45RA^−^) or terminal differentiated effector memory T cells (T_EMRA_, CD62L^−^ CD45RA^+^), cells were stained with peridinin chlorophyll (PerCP)-conjugated anti-CD3, allophycocyanin (APC)-conjugated anti-CD8, fluorescein isothiocyanate (FITC)-conjugated anti-CD19, APC/Cyanin 7 (Cy7)-conjugated anti-CD62L and anti-CD45RA-phycoerythrin (PE)/Cy7 (all BioLegend) and analyzed by multicolor flow cytometry (FACSCanto II, FACSDiva V8.1.2 software, BD Biosciences). Gates were set on the properties regarding light scatter of lymphocytes. At least 50,000 events were acquired in the CD3^+^ gate.

### 4.5. Gene and microRNA Expression Analysis

Total RNA from CD3^+^ T cells isolated from healthy thrombocyte donors (*n* = 5; untreated and SnMP-treated) was isolated using the mirVana RNA Isolation Kit (Thermo Fisher Scientific), and cDNA was reverse transcribed using either the High Capacity cDNA Reverse Transcription Kit or the MicroRNA Transcription Kit (both Thermo Fisher Scientific) according to the manufacturer’s instructions. Expression of IFN-γ and miRNA-155 was quantified using inventoried mixes (Thermo Fisher Scientific). Expression of GAPDH and miRNA-191 served as internal controls.

### 4.6. WT1-Specific T-Cell Stimulation Using HLA-Ig-Based aAPCs

In order to determine the direct effects of SnMP on WT1-specific T cells, artificial antigen-presenting cells (aAPCs) were generated by attaching HLA-A*02:01 molecules (DimerX, Becton Dickinson, Heidelberg, Germany) loaded with HLA*02:01-restricted WT1_37–45_ (VLDFAPPGA, Immudex, Copenhagen, Denmark) and anti-CD28 (BD) monoclonal antibodies (mAbs) to Dynabeads magnetic beads for T-cell stimulation (Life Technologies, Carlsbad, CA, USA), as described previously [66]. PBMCs were isolated from healthy donors (*n* = 5) and allowed to rest overnight. The cells were then stimulated with WT1_37–45_ peptide-loaded aAPCs in the presence and absence of SnMP for 24 h in RPMI1640 + 5% AB serum (T-Cell Medium 2, TCM2) supplemented with 1 µg/mL CD28. For frequency and phenotype analysis, WT1-specific T cells were stained with PE-conjugated dextramer HLA-A*02:01-restricted WT1_37–45_ peptide as well as against CD3, CD8, CD45RA, CD62L and CD19.

On day 0, CD137^+^ cells were isolated using anti-CD137 APCs and anti-APC magnetic beads from Miltenyi Biotec according to the manufacturer’s instructions. CD137-positive CD3^+^, CD4^+^ and CD8^+^ T cells (10^4^ cells / 160 µL per well) were co-cultured with aAPCs at a ratio of 1:1 in TCM2 supplemented with 1% sodium pyruvate, 0.4% MEM Vitamins, 1% non-essential amino acids plus 5 ng/µL IL-7, 5 ng/µL IL-15 and 10 ng/µL IL-21 (all PeproTech) on 96-well round-bottom plates. On day 3 or 4, another 80 µL of the medium was added. On day 7, cells were harvested and IFN-γ and granzyme B ELISpot assays and ELISAs were performed.

### 4.7. Analyzing Cell Culture Supernatants via IFN-γ and Granzyme B ELISA

The amount of secreted IFN-γ or granzyme B in the supernatant formed during the expansion of WT1-specific T cells with and without SnMP-based HO-1 inhibition in an antigen-independent setting was determined using the Human IFN gamma ELISA Ready-SET-Go!^®^ Kit and the Granzyme B Human ELISA Kit (both ThermoFisher Scientific) according to the manufacturer’s instructions. Read-out was performed by using the ELISA reader Synergy 2 (BioTek).

### 4.8. Cell Activation After ppWT Stimulation, as Determined via IFN-γ ELISpot

The effect of SnMP-based HO-1 inhibition on the functionality of WT1-specific T cells was analyzed by IFN-γ ELISpot assay, as described previously [67,68]. Briefly, freshly isolated PBMCs were resuspended at a concentration of 1 × 10^7^ cells/mL in TCM1 and allowed to rest overnight on a 24-well plate (37 °C, 5% CO_2_). On day 1, PBMCs were washed (300× *g*, 10 min, RT), resuspended in TCM1, counted, seeded at a concentration of 2.5 × 10^5^ cells/well in a 96 well plate (MultiScreen HTS IP Sterile Plate; Millipore, Darmstadt, Germany) and stimulated overnight with 2.5 µg/mL CEF peptide pool (PANATecs, Heidelberg, Germany; positive control), ppWT1 (1 µg/mL pro peptide) with or without SnMP, and DMSO (solvent control), respectively. Untreated cells served as the negative control. ELISpot was performed on the next day. Spots were counted using the ImmunoScan Core Analyzer, and the results were analyzed using ImmunoSpot 5.0 Academic software (both from Cellular Technology Ltd. Bonn, Germany). Spot counts at least two times higher than the spot count of the negative control were regarded as positive. ELISpot analyses were performed using monoclonal IFN-γ antibody 1-D1K (coating), 7-B6-1-Biotin (detection) and streptavidin obtained from Mabtech (Sweden), and NBT/BCIP substrate solutions (development) from Serva (Heidelberg, Germany).

### 4.9. Flow Cytometric Assessment of Phosphorylated ERK

To demonstrate the effect of HO-1 inhibition on intracellular signaling pathways, PBMCs were stimulated using ppWT1 with or without SnMP for 15 min at 37 °C. PBMCs cultured in PMA (50 ng/mL) served as the positive control. Intracellular staining, fixation and permeabilization of the cells were done by using Fix Buffer I and Perm Buffer III following the manufacturer’s instructions (BD). Phosphorylated ERK (pERK) expression was assessed by flow cytometry after staining for pERK1/2 (BD), CD3^+^, and CD8^+^ cells. An IgG1 isotype control was analyzed additionally in each experimental set-up.

### 4.10. Use of SnMP to Enhance the Isolation of WT1-Specific T Cells in a Near-Clinical Setting

An IFN-γ CSA (IFN-γ Secretion Assay–Cell Enrichment and Detection Kit, Miltenyi Biotec) was used to evaluate the effect of SnMP on the enrichment of WT1-specific T cells. Freshly isolated PBMCs were resuspended at a concentration of 1 × 10^7^ cells/mL in TexMACS GMP Medium (Miltenyi Biotec) and allowed to rest overnight. WT1-specific T cells were enriched by using the CSA to mimic a situation very comparable to a clinical setting. PBMCs were stimulated with ppWT1 in the presence and absence of SnMP overnight. This was followed by the detection of IFN-γ secreting cells and their enrichment according to the manufacturer’s instructions.

For flow cytometric analysis, cells were stained before and after enrichment with mAbs against CD3, CD8, CD56 and CD45. Dead cells were excluded by using 7-amino-actinomycin D (7-AAD). Gates were based on the light scatter properties of lymphocytes and on CD3^+^/IFN-γ^+^ cell populations. The PE-conjugated anti-IFN-γ antibody was obtained from Miltenyi Biotec.

For differentiation between antiviral and WT1-specific T-cell responses, PBMCs were stimulated overnight with pools of EBV-derived (1 µg/mL pro peptide) and WT1-derived peptides with or without SnMP. The IFN-γ CSA was performed as described.

### 4.11. Effect of SnMP on the Expression of CD137

To determine CD137 expression on activated T cells, freshly isolated PBMCs from six donors were isolated, rested overnight in RPMI1640 + 5% AB serum (TCM2), and then stimulated with either ppWT1, pWT1_37–45_ or pWT1_126–134_ peptides in the presence or absence of SnMP for 24 h. Peptide-stimulated T cells were stained with PE-conjugated dextramer HLA-A*02:01-restricted to pWT1_37–45_ or pWT1_126–134_ and surface-labeled for Abs against CD3, CD8, CD137, CD45RA and CD62L.

### 4.12. ELISpot for Detection of Granzyme B Release After Expansion with aAPCs

The specific effector function of the pWT1_37–45_ expanded T cells was assessed by measuring their granzyme B release when incubated with corresponding target cells. K562^+^A*02 target cells were loaded with the HLA*02:01-restricted WT1 peptides pWT1_37–45_ and pWT1_126–134_ (10 µg/mL on day 6, overnight), which were added on day 7 to the WT1-specific T cells (effector cells) and incubated overnight. Effector (E) and target (T) cells were incubated at a ratio of 0.01:1, 0.05:1 and 0.1:1 and the Granzyme B ELISpot assay was performed as described previously [67].

### 4.13. Statistical Analysis

The data were analyzed using the paired *t* test and the two-way ANOVA, which were run on GraphPad PRISM V5.01 software (GraphPad Software, San Diego, CA). Regarding significance levels, *p* values ≤ 0.05 were defined as significant (*), and *p* ≤ 0.01 as very significant (**).

## Figures and Tables

**Figure 1 ijms-20-00482-f001:**
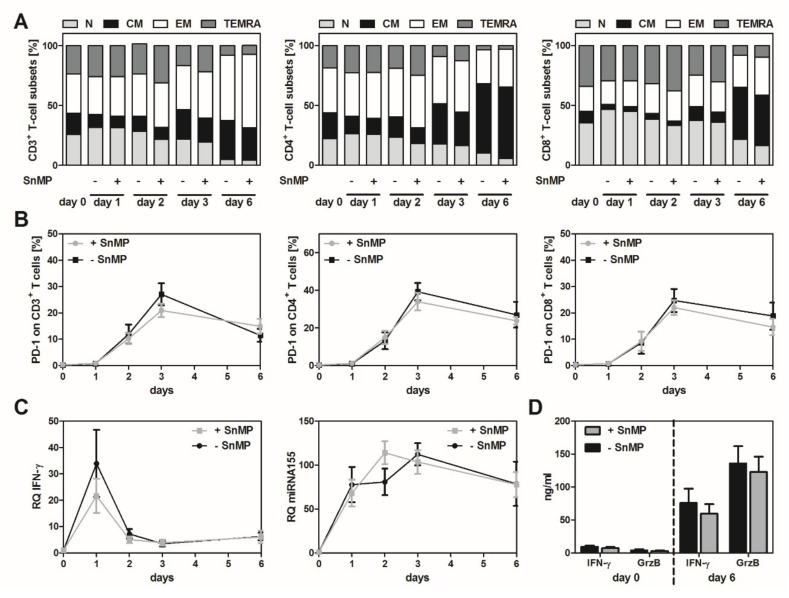
Effect of heme oxygenase-1 (HO-1) inhibition in an antigen-independent setting. CD3^+^ T cells were isolated from peripheral blood mononuclear cells (PBMCs) from seven healthy donors and stimulated with CD3/CD28 Dynabeads^®^ for six days with or without tin mesoporphyrin (SnMP) (10 µM). On days 1, 2, 3, and 6, cells and supernatants were acquired for analysis. (**A**) No significant change in the composition of T-cell subsets was observed in the CD3^+^, CD4^+^, and CD8^+^ T-cell populations. Data represent the means of seven donors. (**B**) PD-1 expression did not change significantly in the presence or absence of SnMP in the CD3^+^, CD8^+^ and CD4^+^ T-cell populations. There was no significant difference between the SnMP-treated and SnMP-untreated cells in the CD3^+^, CD8^+^ or CD4^+^ T-cell populations. Data represent the means of seven donors. (**C**) mRNA levels of IFN-γ and miRNA-155 were analyzed by real-time PCR. Data represent the means of five donors. (**D**) ELISAs performed to assess the amount of granzyme B and IFN-γ in the supernatant showed no significant difference in the amount of IFN-γ or granzyme B in cells treated with or without HO-1 inhibition via SnMP. Data represent the means of seven donors.

**Figure 2 ijms-20-00482-f002:**
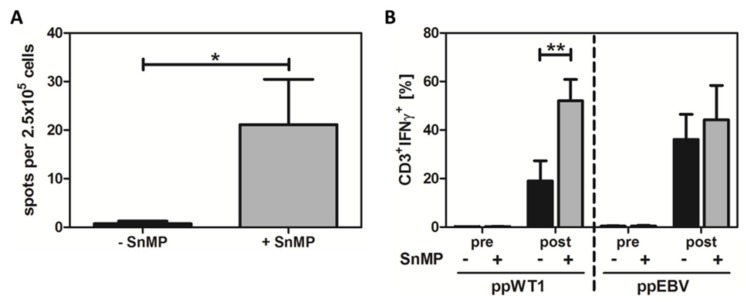
SnMP significantly enhanced T-cell responses to Wilms tumor protein-1 (WT1) stimulation and increased the amounts of antiviral and WT1-specific IFN-γ^+^ T cells. (**A**) IFN-γ ELISpot was used to measure immune responses in 50 healthy donors stimulated using ppWT1. Thirteen (26%) donors showed a positive response of IFN-γ-positive T cells to stimulation with ppWT1, which increased 30.1-fold after HO-1 inhibition with SnMP. Data were normalized to the controls and are presented as mean ± SEM of 13 experiments. Single asterisk (*) indicates *p* = 0.0478 (two-tailed paired t test). (**B**) PBMCs from the donors were isolated and stimulated with ppWT1 (left) or ppEBV (right) in the presence or absence of HO-1 inhibitor SnMP. Prior to enrichment, ppWT + SnMP-treated cells displayed a higher (1.2-fold) response to ppWT1 than those treated with ppWT1 alone. After enrichment, the amount of CD3^+^/IFN-γ^+^ T cells was significantly higher (2.7-fold) in ppWT1 + SnMP-treated cells. Data are presented as the mean ± SEM of 9 experiments. Double asterisk (**) indicates *p* = 0.01 (two-way ANOVA). Although there was no significant effect of stimulation with ppEBV, there was a positive tendency towards enrichment of IFN-γ-positive CD3^+^ T cells. After HO-1 inhibition via SnMP, the number of T cells increased around 1.2-fold compared to the numbers in untreated cells before and after enrichment. Data represent the mean ± SEM of 4 experiments. ppWT1, pool of peptides derived from WT1; ppEBV, pool of peptides derived from Epstein-Barr virus.

**Figure 3 ijms-20-00482-f003:**
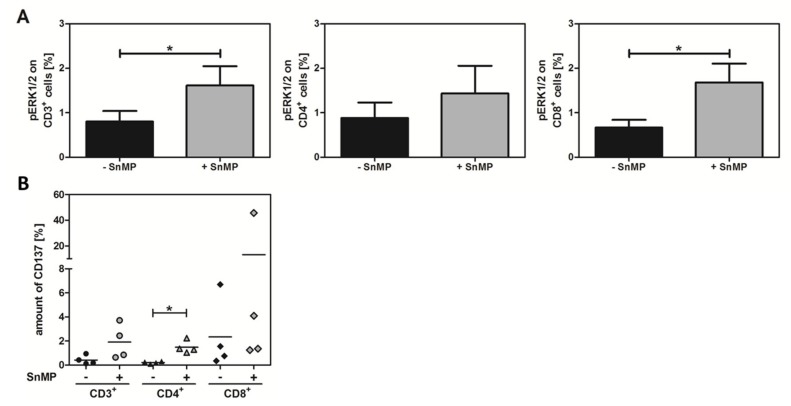
Significant effect of SnMP on phosphorylated extracellular signal-regulated kinase (pERK) and CD137. (**A**) PBMCs were isolated and stimulated using ppWT1 with or without SnMP for 15 min. After fixation and permeabilization, the percentage of pERK1/2 on CD3^+^, CD8^+^, and CD4^+^ T cells was assessed by flow cytometry. The amount of pERK1/2 on the target cells increased after stimulation with ppWT1+SnMP compared to stimulation with ppWT1 alone was observed in the CD3^+^ (2-fold), CD4^+^ (1.6-fold), and CD8^+^ (2.5-fold) T-cell populations. Data are presented as the mean ± SEM of six independent experiments. Single asterisk (*) indicates *p* ± 0.0226 (two-tailed paired t test). (**B**) Higher expression of CD137 was observed in CD3^+^, CD4^+^, and CD8^+^ T cells after stimulation with ppWT1+SnMP compared to ppWT1 alone. Data represent the means of four independent experiments. Single asterisk (*) indicates *p* = 0.0147 (two-way ANOVA). ppWT1, pool of peptides derived from WT1.

**Figure 4 ijms-20-00482-f004:**
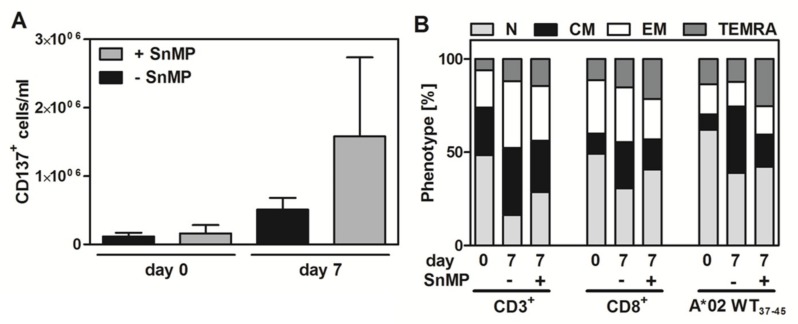
Effect of SnMP on WT1 peptide-specific T-cell expansion. (**A**) The increase in the total cell count was 1.4-fold (day 0) and 3.1-fold (day 7) higher in cells treated with SnMP+pWT1_37–45_ than in those treated with pWT1_37–45_ alone. Data are presented as the mean ± SEM of four experiments. (**B**) The composition of T-cell subsets did not change significantly in the CD3^+^, CD8^+^ and A*02 WT1_37–45_ stained T-cell populations. Values represent the means of five donors.

**Figure 5 ijms-20-00482-f005:**
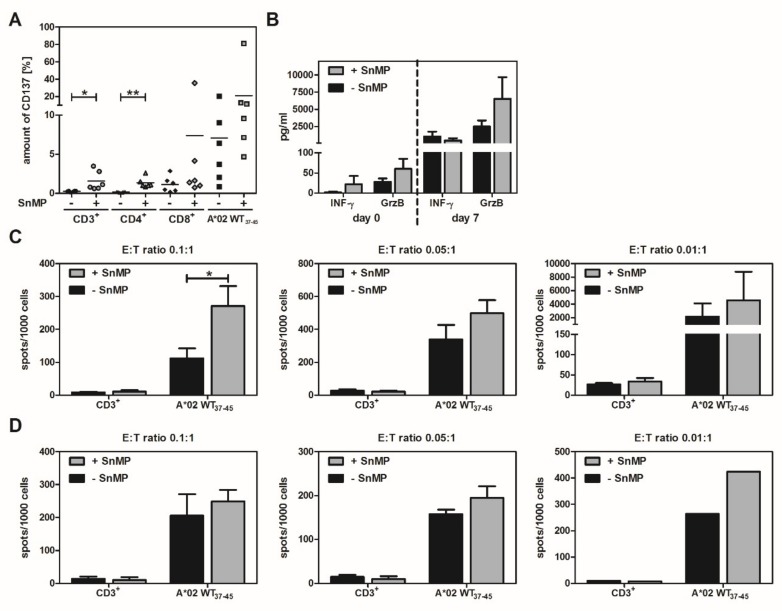
Effect of SnMP on WT1 peptide-specific T-cell functionality. (**A**) Higher expression of CD137 was observed on CD3^+^, CD4^+^, and CD8^+^ T cells from PBMCs treated with SnMP after stimulation with the specific peptide pWT1_37–45_. Data represent the mean of six independent experiments. Single asterisk (*) indicates *p* = 0.0430, and double asterisk (**) indicates *p* = 0.0075 (two-way ANOVA). (**B**) The IFN-γ ELISA revealed a similar concentration of secreted IFN-γ in the supernatant on day 7 in the group treated with pWT1_37–45_ + SnMP compared to pWT1_37–45_ peptides alone (*n* = 5). The corresponding granzyme B ELISA showed a higher amount of granzyme B after treatment the cells with SnMP on days 0 and 7 than without SnMP (*n* = 4). On day 7, the expanded cells were incubated with peptide-loaded target cells at different effector/target (E:T) ratios. Granzyme B (**C**) and IFN-γ (**D**) secreting cells were detected by ELISpot. Granzyme B ELISpot data are presented as the mean ± SEM of 3 independent experiments (ratio of 0.1:1 and 0.05:1) and as mean ± S.D with *n* = 2 for a ratio of 0.01:1. IFN-γ ELISpot data are presented as mean ± standard deviation (SD) for 2 donors (effector:target (E:T) ratio: 0.1:1), 3 donors (E:T ratio: 0.05:1), and one donor (E:T ratio: 0.01:1). Single asterisk (*) indicates *p* = 0.0441 (two-way ANOVA).

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
