# Peer review of "Inhibition of Heme Oxygenase-1 Activity Enhances Wilms Tumor-1-Specific T-Cell Responses in Cancer Immunotherapy"

_ijms, 2019, doi:10.3390/ijms20030482_

Reviewer 1 Report

Schillingmann et al suggest that inhibition of HO-1 can enhance WT1-specific T cell responses and this can be applicable for anti-cancer immune therapy. Ho-1 is known to involve in regulation of T cell responses through direct regulation of T cell activation and APC regulations. In their data, SnMP can regulate T cell responses. However, SnMP did not dramatically regulate anti-CD3/CD28-activated T cell responses while WT1-specific T cell responses were regulated by SnMP. The study is well organized and experimental data support their conclusion. However, there are several questions about the data.

Issue #1: In figure 1, for checking of T cell activation, the author also need to check IL-2 secretion and proliferation of activated T cells.

Issue #2: SnMP did not dramatically regulate anti-CD3/CD28-activated T cell responses while WT1-specific T cell responses were regulated by SnMP. Did you test other antigen-specific T cell responses? Possibly, SnMP more strongly regulates APCs than T cells because anti-CD3/CD28-activated T cell responses were not dramatically regulated by SnMP. Have you checked APC regulation by SnMP?

Issue #3: WT1 specific T cells are present in healthy people. Possibly, WT1-specific T cell responses are normally inhibited by a general immune suppression mechanism because WT1 is an autoantigen in healthy people. Have checked the characteristics of WT1-specific T cells such as PD-1 expressions.

Issue #4: in figure 5A, what are the % of CD137+ cells in the CD4+ A*02 WT37-45-(negative), CD4+ A*02 WT37-45-(negative), and CD8+ A*02 WT37-45-(negative) cells under the conditions? Are the expression of CD137+ on non-WT1 specific T cells increased in the condition? 

Minor issues:

#1. Please use the upper script for “+”.

#2. In line 99, what is a meaning of “the same range”?

#3. In figure 3B, “CD3”, “CD4”, and “CD8” are “CD3+”, “CD4+”, “CD8+”?

#4. In figure 5A, “amount of CD137 [%]” is “% of CD137+ cells”?

Author Response

Dear Editor, dear Reviewer 1,

please find enclosed the revised version of our manuscript “Inhibition of heme oxygenase-1 activity enhances Wilms tumor-1-specific T-cell responses in cancer immunotherapy”, which we have carefully edited according to the reviewers’ comments. We would like to thank the reviewer for his helpful comments, which we think have significantly improved the quality of the manuscript.

Please find answers to all questions below. For your convenience, the original question (issue) is always cited above the response. Furthermore, we highlighted changes in the manuscript.

With best regards,

Damaris A. Schillingmann, Sebastian B. Riese, Vijith Vijayan, Sabine Tischer, Helga Schmetzer, Britta Maecker-Kolhoff, Rainer Blasczyk, Stephan Immenschuh, Britta Eiz-Vesper

Response to Reviewer 1

Schillingmann et al. suggest that inhibition of HO-1 can enhance WT1-specific T cell responses and this can be applicable for anti-cancer immune therapy. Ho-1 is known to involve in regulation of T cell responses through direct regulation of T cell activation and APC regulations. In their data, SnMP can regulate T cell responses. However, SnMP did not dramatically regulate anti-CD3/CD28-activated T cell responses while WT1-specific T cell responses were regulated by SnMP. The study is well organized and experimental data support their conclusion. However, there are several questions about the data.

Issue #1: In figure 1, for checking of T cell activation, the authors also need to check IL-2 secretion and proliferation of activated T cells.

Response 1: We agree with the reviewer that proliferation rate and cytokine secretion pattern are important to determine the influence of tin mesoporphyrin (SnMP) on T cells. Therefore, we addressed these major points in Figures 1 and 4. The influence of HO-1 modification in an antigen-independent setting was described by the work of Burt et al., and a significantly increased proliferation of CD4+ and CD8+ T cells after treatment of peripheral blood mononuclear cells (PBMCs) with SnMP compared to untreated cells .[1].

(see introduction page 2, line 64-68 and line 70-72)

Addressing the matter of proliferation of SnMP treated T cells, we showed that total cell count of CD137+ T cells in PBMCs treated with pWT137–45 in the presence and absence of SnMP increased slightly after a seven-day-culture (Fig. 4A). Bunse, Fortmeier et al., too, showed an increased proliferation of T cells following antigen stimulation and SnMP treatment via measuring carboxyfluorescein succinimidyl ester (CFSE) staining [2]

IL-2 is an important growth factor and was used in the culture medium (see materials and methods). Since this would alter the corresponding results we decided to not determine the level of IL-2 secretion and instead determined the secretion of IFN-γ and granzyme B via ELISA (Fig. 1D) and the RNA level of IFN-γ and miRNA-155 instead (Fig. 1C).

However, other cytokines and mediators could be examined to further characterize the effect of inhibition of heme oxygenase-1 (HO-1) activity via SnMP. Kapturczak et al. showed an increase of IL-6, IL-10 and TNFα in splenocytes of HO-1 knock-out mice using multiplex immunoassays [3]. Bunse, Fortmeier et al. obtained similar results after SnMP-treatment of peptide-stimulated PBMCs in vitro by multiplex technique [2].

(see Figure 1 and discussion page 8, line 197198 and line 277279)

[1] Burt, T. D. et al., Naive human T cells are activated and proliferate in response to the heme oxygenase-1 inhibitor tin mesoporphyrin. J Immunol 2010, 185, (9), 5279–88

[2] Bunse, C. E., Fortmeier, V. et al., Modulation of heme oxygenase-1 by metalloporphyrins increases anti-viral T cell responses. Clin Exp Immunol 2015, 179, (2), 265–76

[3] Kapturczak, MH et al., Heme oxygenase-1 modulates early inflammatory responses: evidence from the heme oxygenase-1-deficient mouse. Am J Pathol 2004, 165, 1045–53

Issue #2: SnMP did not dramatically regulate anti-CD3/CD28-activated T cell responses while WT1-specific T cell responses were regulated by SnMP. Did you test other antigen-specific T cell responses? Possibly, SnMP more strongly regulates APCs than T cells because anti-CD3/CD28-activated T cell responses were not dramatically regulated by SnMP. Have you checked APC regulation by SnMP?

Response 2: We thank the reviewer for the interesting remark and are pleased to address the question regarding the initiation of other antigen-specific T-cell responses and the regulation of antigen-presenting cells (APCs).

We would like to refer to a previous project in our group. Bunse, Fortmeier et al., found that the modulation of the enzymatic activity of HO-1 increased the activity and proliferation of human Cytomegalovirus (CMV)-specific T cells significantly.

In comparison to the stimulation with the tumor-associated antigen ppWT1, we also tested the influence of HO-1 modulation via SnMP in direct comparison to stimulation with the strong viral antigen, ppEBV Consensus (peptide pool derived from Epstein-Barr virus), which is shown in Figure 2B. After enrichment we got a 1.2fold higher response of the CD3+/INF-γ+ cells with SnMP than without SnMP. However, the results in this case were not significant.

Addressing the question concerning APC regulation, we have not directly addressed this issue in the current work because we have used aAPCs (artificial APCs). However, in our previous study we did not observe any significant effect of SnMP treatment on APC maturation, function and T cell stimulatory capacity [1].

[1] Bunse, C. E., Fortmeier, V. et al., Modulation of heme oxygenase-1 by metalloporphyrins increases anti-viral T cell responses. Clin Exp Immunol 2015, 179, (2), 265–76

Issue #3: WT1 specific T cells are present in healthy people. Possibly, WT1-specific T cell responses are normally inhibited by a general immune suppression mechanism because WT1 is an autoantigen in healthy people. Have checked the characteristics of WT1-specific T cells such as PD-1 expressions.

Response 3: We thank the reviewer for this helpful and important point and discussed now the characterization of WT1-specific T cells in healthy donors in more detail.

The expression levels of programmed cell death protein 1 (PD-1) on Wilms tumor-1 (WT1)-specific T cells were not studied, but we analyzed the expression on the antigen-independent stimulation (Fig. 1B). Here we found, that SnMP had no significant effect on the expression of PD-1 in CD3+, CD8+ and CD4+ T-cell populations. The highest PD-1 expression levels were found on day 3 for all T-cell populations. Interestingly, the PD-1 expression in SnMP-treated cells was 3% to 6% lower than in SnMP-untreated cells (Fig. 1B). PD-1 is an important subject in the field of immunotherapy, as it is already used as a target in antibody therapy in the treatment of metastatic melanoma [1] and ongoing research is performed on its role in adoptive T-cell transfer. A transient upregulation of PD-1 and other exhaustion markers may be detected in WT1-specific T cells obtained from peripheral blood of healthy donors upon restimulation with the antigen, as this is also used to describe T cell activation [2]. This was also shown by Bak et al. in the case of CMV-specific T cells [3]. It was recently examined that tumor-infiltrating T cells specific for myeloma and melanoma in mice, selected for their expression of PD-1 exhibited a superior tumor control and specificity, which was discussed by Simon and Labarriere in their review [4]. If the activity of WT-1-specific T cells in healthy individuals was in part influenced by the expression of PD-1 under the influence of SnMP, this circumstance might be used to further increase functionality and avidity of those cells used in adoptive T-cell transfer.

(see Figure 1 and discussion page 8, line 226–229)

[1] Bhandaru M and Rotte A., Monoclonal Antibodies for the Treatment of Melanoma: Present and Future Strategies. Methods Mol Biol 2019, 1904, 83–108

[2] Catakovic K, Klieser E, Neureiter D, Geisberger R. T cell exhaustion: from pathophysiological basics to tumor immunotherapy. Cell communication and signaling: CCS 2017 15(1):1. doi:10.1186/s12964-016-0160-z

[3] Bak, S., Tischer, S., Dragon, A., Ravens, S., Pape, L., Koenecke, C., Eiz-Vesper, B. (2018). Selective Effects of mTOR Inhibitor Sirolimus on Naïve and CMV-Specific T Cells Extending Its Applicable Range Beyond Immunosuppression. Frontiers in Immunology 2018 9, 2953. https://doi.org/10.3389/fimmu.2018.02953

[4] Simon S. and Labarriere N., PD-1 expression on tumor-specific T cells: Friend or foe for immunotherapy? OncoImmunology 2018, 7, (1), e1364828

Issue #4: in figure 5A, what are the % of CD137+ cells in the CD4+ A*02 WT37-45-(negative), CD4+ A*02 WT37-45-(negative), and CD8+ A*02 WT37-45-(negative) cells under the conditions? Are the expression of CD137+ on non-WT1 specific T cells increased in the condition?

Response 4: We thank the reviewer for drawing our attention to the influence of SnMP on non-WT1-specific T cells regarding the expression of CD137 as an activation marker. pMHC class I multimers bind to the TCR on CD8 T cells and therefore only antigen-specific CD8+ T cells become detectable. Furthermore the peptide WT3745 is HLA-A*02:01-restricted and therefore only activates CD8+ T cells. Due to these facts the detection of multimer-positive CD4+ T cells is not possible.

In the reanalysis of the CD8+/multimer T-cell population we determined only slight upregulation of CD137 and therefore activation (pWT137-45 peptide only: 1.1%, pWT13745+SnMP treatment: 7.4%). In contrast in the specific CD8+/multimer+ fraction we determined an increase of CD137 from 7.1% (pWT13745 peptide alone) to 21.1% (pWT13745+SnMP).

(see Fig. 5A and line 173181)

Issue #5, minor issues:

#1. Please use the upper script for “+”.

Response: According to the appreciated suggestions of the reviewer, we revised the manuscript and used now the upper script.

#2. In line 99, what is a meaning of “the same range”?

Response: We also thank the reviewer for pointing out this fact. We revised the section in the manuscript to avoid a possible misunderstanding.

(see results page 3, line 102)

#3. In figure 3B, “CD3”, “CD4”, and “CD8” are “CD3+”, “CD4+”, “CD8+”?

Response: Yes, they are. We revised the Figure 5B.

#4. In figure 5A, “amount of CD137 [%]” is “% of CD137+ cells”?

Response: The “amount of CD137 [%]” means the amount of CD137+ on CD3+ , CD4+ as well as CD8+ T cells as indicated on the y-axis.

Reviewer 2 Report

The manuscript entitled " Inhibition of heme oxygenase-a activity enhances Wilms tumor-1-specific T-cell response in cancer immunotherapy ", prepared by Damaris Schillingmann, Sebastian Riese, et. al. is an interesting and good prepared paper with subject of T-cell therapy for cancer treatment.   
The subject of this work is good fit to the scope of International Journal of Molecular Sciences, and the results reported in the manuscript should demonstrate the benefits of the research for cancer treatment, especially the field of immunotherapy. In my opinion, there is some novelty, potentially interested for international scientific audiences on the immunotherapy of cancer through upregulating heme oxygenase-1 activities. However, the research data showing should be improved, and some data need to be explained more reasonably. Since then, I am suggesting to be accepted after major revision by International Journal of Molecular Sciences. More raw data should be added before publishing. Scientific paper cannot only contain statistical results, it should show experimental raw data.

Basically, the paper reported project tried to regulate the heme oxygenase activity through the inhibitor, SnMP, and the related in vitro testing was performed to prove the hypothesis that the SnMP could regulate related enzyme and protein expression level, so that it could successfully inhibit HO-1 functional, which benefit the specific T-cell activation and proliferation. Actually, this is a fundamental research project for immunotherapy which could guide the developed T-cell immunotherapy for caner. Although, the authors mentioned several times of “Clinical applicability”, there was only in vitro testing. How about the toxicity of SnMP for the patient, how could the SnMP be delivered in vivo, what is the side effect, if the induced inhibition of HO-1 will affect the regular functions of T-cell? Without the preliminary in vivo data, the hypothesis is hard to be proved.

Even for the in vitro testing only:

The basic information of patient, who donated the testing samples, should be given.

Basically, the protein level, like pERK1/2, should be tested by western blot. The related RNA level should be determined by qPCR. Why the authors used the flow cytometry? Even for the flow cytometry, the raw figures should be supplied, not only the statistical results. In addition, the project needs at least two strategies to prove the protein level, not only flow cytometry.

In addition, the figure 5 is in advanced of figure 4 in the manuscript.        

Author Response

Dear Editor, dear Reviewer 2,

please find enclosed the revised version of our manuscript “Inhibition of heme oxygenase-1 activity enhances Wilms tumor-1-specific T-cell responses in cancer immunotherapy”, which we have carefully edited according to the reviewers’ comments. We would like to thank the reviewer for his helpful comments, which we think have significantly improved the quality of the manuscript.

Please find answers to all questions below. For your convenience, the original question (issue) is always cited above the response. Furthermore, we highlighted changes in the manuscript.

With best regards,

Damaris A. Schillingmann, Sebastian B. Riese, Vijith Vijayan, Sabine Tischer, Helga Schmetzer, Britta Maecker-Kolhoff, Rainer Blasczyk, Stephan Immenschuh, Britta Eiz-Vesper

Response to Reviewer 2

The manuscript entitled "Inhibition of heme oxygenase-a activity enhances Wilms tumor-1-specific T-cell response in cancer immunotherapy ", prepared by Damaris Schillingmann, Sebastian Riese, et. al. is an interesting and good prepared paper with subject of T-cell therapy for cancer treatment.

The subject of this work is good fit to the scope of International Journal of Molecular Sciences, and the results reported in the manuscript should demonstrate the benefits of the research for cancer treatment, especially the field of immunotherapy. In my opinion, there is some novelty, potentially interested for international scientific audiences on the immunotherapy of cancer through upregulating heme oxygenase-1 activities. However, the research data showing should be improved, and some data need to be explained more reasonably. Since then, I am suggesting to be accepted after major revision by International Journal of Molecular Sciences. More raw data should be added before publishing. Scientific paper cannot only contain statistical results, it should show experimental raw data.

Basically, the paper reported project tried to regulate the heme oxygenase activity through the inhibitor, SnMP, and the related in vitro testing was performed to prove the hypothesis that the SnMP could regulate related enzyme and protein expression level, so that it could successfully inhibit HO-1 functional, which benefit the specific T-cell activation and proliferation. Actually, this is a fundamental research project for immunotherapy which could guide the developed T-cell immunotherapy for caner. Although, the authors mentioned several times of “Clinical applicability”, there was only in vitro testing. How about the toxicity of SnMP for the patient, how could the SnMP be delivered in vivo, what is the side effect, if the induced inhibition of HO-1 will affect the regular functions of T-cell? Without the preliminary in vivo data, the hypothesis is hard to be proved.

Even for the in vitro testing only:

The basic information of patient, who donated the testing samples, should be given.

Basically, the protein level, like pERK1/2, should be tested by western blot. The related RNA level should be determined by qPCR. Why the authors used the flow cytometry? Even for the flow cytometry, the raw figures should be supplied, not only the statistical results. In addition, the project needs at least two strategies to prove the protein level, not only flow cytometry. In addition, the figure 5 is in advanced of figure 4 in the manuscript.

Issue #1: More raw data should be added before publishing. Scientific paper cannot only contain statistical results, it should show experimental raw data.

Response 1: We thank the reviewer for that suggestion. To support the findings of our manuscript we have added as an example the IFN-γ ELISpot for one donor with and without tin mesoporphyrin (SnMP) treatment and after stimulation with ppWT1 (overlapping peptide pool of peptides derived from WT1) within the supplementary data (Supplementary Figure (SF) SF1). In addition, we performed pERK1/2 detection via Western blot (see issue #4) and included a FACS plot of the pERK1/2 detection (SF2).To underline the effect of SnMP on T cells after stimulation with the A*02-restricted pWT137−45, we show an exemplary FACS plot (SF3) of CD137+/CD8+ T cells, which were labeled with the A*02-restricted pWT137−45 multimer.

Issue #2: There was only in vitro testing. How about the toxicity of SnMP for the patient, how could the SnMP be delivered in vivo, what is the side effect, if the induced inhibition of HO-1 will affect the regular functions of T-cell? Without the preliminary in vivo data, the hypothesis is hard to be proved.

Response 2: The use of SnMP in the manufacturing of tumor-specific T cells will be done using the IFN-g cytokine capture system (CCS) and the CliniMACS Prodigy device (Priesner et al., 2016; discussion, page 8, lines 193-196). In this process, PBMCs would be obtained from healthy donors and restimulated with the respective antigens in the presence of SnMP in vitro. Following a huge number of wash steps in order to remove unspecific T cells and peptides as well as SnMP, purified WT1-specific T cells would be transferred to the patient. Therefore SnMP itself would not be applied directly to either the patient or the donor. This procedure significantly limits potential side effects and alterations to the patient’s own T cells.

To address the usage of SnMP in clinical trials, we have extended the discussion part. Interestingly, a phase 1 clinical study on RBT-1, a SnPP derived product planned to be used in patients with chronic kidney injury, is currently in preparation and registered (see https://clinicaltrials.gov/ct2/show/NCT03630029 for further information). Moreover, there are case studies on the use of SnMP for treating hyperbilirubinemia in infants.

(see discussion, page 8, lines 193-196, page 9, line 293-295)

Issue #3: The basic information of patient, who donated the testing samples, should be given.

Response 3: We thank the reviewer for the remark and would like to draw attention to the corresponding paragraph in materials and methods (page 10, lines 319326). Donors were healthy platelet donors with no prior history of blood transfusion and acute infection. Donor age ranged from 24 years to 65 years. Approximately 75 % of the tested WT1-positive donors are male individuals.

Issue #4: Basically, the protein level, like pERK1/2, should be tested by western blot. The related RNA level should be determined by qPCR. Why the authors used the flow cytometry? [] In addition, the project needs at least two strategies to prove the protein level, not only flow cytometry.

Response 4: Multicolor flow cytometric evaluation of the phosphorylation level of ERK was done because it was shown to be a very sensitive and fast method, which allows for the detection of pERK in specific cells subsets in parallel. Raw data were now included (SF2).

According to the reviewer suggestion we performed now western blot analysis for pERK1/2. For this, CD3+ T cells were stimulated with ppWT1 and incubated with or without SnMP for 15 min at 37 °C. The samples were washed with PBS and lysed with RIPA buffer for 30 min on ice. Protein concentration was determined via BCA; subsequently SDS-PAGE and Western blot were performed. For immunodetection we used an anti-pERK1/2 antibody (Cell Signaling Technology) followed by staining with the secondary anti-mouse antibody. For detection of total ERK we used an anti-ERK antibody (Cell Signaling Technology) and an anti-rabbit antibody as secondary antibody. Analysis was done with the ChemiDoc Imaging System (Bio-Rad).

Find below the resulting image:

We analyzed three different donors and used different amounts of protein for the SDS-PAGE (donor 1: 22 µg, donor 2: 11 µg and donor 3: 14 µg). For donor 1 the characteristic double band for pERK1/2 is well detectable (blot on top). In the other donors the double band is weaker. The band for pERK1/2 in the untreated control of donor 1 is weaker compared to the ppWT1 and ppWT1 + SnMP treated samples. Furthermore it seemed, that the double band in the sample ppWT1 + SnMP is a little bit more prominent compared to the sample with ppWT1 alone. The blot on the bottom showed the detection of the total ERK amount on the same blot.

Taken together, our preliminary results also by Western Blot analysis confirmed the increase of pERK1/2 in CD3+ T cells after stimulation with ppWT1 in the presence of SnMP compared to the absence of SnMP.

Issue #5: In addition, the figure 5 is in advanced of figure 4 in the manuscript.

Response 5: We thank the reviewer for drawing our attention to this point. We have changed the manuscript accordingly.

(see line 153–161 to 173–180) 

Round  2

Reviewer 1 Report

All issues are addressed.

Reviewer 2 Report

Thanks for the authors' reply to my comments. I basically agree with the authors' points and supplied data, and suggested to accept the paper in present form.